# CONCEPT-GUIDED TOKENIZATION: CLOSING THE GAP BETWEEN RECONSTRUCTION AND GENERATION

## ABSTRACT

Recent advances in image generation have been largely driven by image tokenization, which compresses raw pixels into compact latent representations. While existing tokenizers excel at preserving low-level visual details through reconstruction-based training, they often lack explicit semantic guidance, which limits their ability to capture semantically structured representations and thus hinders their performance on downstream tasks like image generation. To overcome this limitation, we propose a novel tokenization framework that incorporates high-level semantics through two key innovations: (1) a text-integrated encoder that jointly processes images and textual descriptions to produce semantically enriched latent representations, and (2) a concept-guided training objective that leverages sparse autoencoders to decompose pre-trained vision–language model features to a semantic concept space, employing sparse and disentangled concept indices for guidance. Our approach achieves stronger alignment with semantic concepts, consequently maintaining high reconstruction fidelity while achieving more competitive downstream image generation performance. By infusing high-level semantic structures into low-level visual fidelity, our method bridges the reconstruction-generation divide and drives generative modeling as a powerful foundation.

## 1 INTRODUCTION

In recent years, image generation has achieved remarkable progress, with diffusion (Rombach et al., 2022; Peebles & Xie, 2023; Hatamizadeh et al., 2024; Shin et al., 2025) and autoregressive (Sun et al., 2024; Li et al., 2024; Tian et al., 2024; Li et al., 2025) models achieving high-quality synthesis. A key enabler of this progress is image tokenization, which compresses raw pixels into a compact latent space (Xiong et al., 2025a). These latent representations, whether continuous (Kingma & Welling, 2014; Li et al., 2024) or discrete (Van Den Oord et al., 2017; Yu et al., 2022), provide an expressive yet computationally efficient alternative to the high-dimensional image space. Tokenization enables generative models to operate directly in the latent domain, simultaneously improving efficiency and synthesis fidelity (Zha et al., 2025; Kim et al., 2025) and thus establishing it as an important component of image generation systems.

Reconstruction-based training (Yu et al., 2024a; Zha et al., 2025; Kim et al., 2025) serves as a primary objective for learning visual tokenizers, as it effectively preserves low-level image details. However, while achieving strong reconstruction performance, this approach often lacks high-level semantic guidance (Qu et al., 2025; Wu et al., 2025b; Zhao et al., 2025), leading to poor generalization in downstream generation tasks and limiting the quality of generated images (Xiong et al., 2025b). To overcome this, existing methods (Qu et al., 2025; Xiong et al., 2025b) typically align tokenizer features with high-level representations from pre-trained vision models such as CLIP (Radford et al., 2021) or DINOv2 (Oquab et al., 2024), or encourage text-image alignment using paired captions (Ge et al., 2024; Liang et al., 2024; Wu et al., 2025b). Yet, direct alignment with pre-trained feature representations introduces both optimization difficulties and generalization challenges. The high dimensionality of these features makes alignment a difficult regression task, where the curse of dimensionality flattens distance metrics and weakens gradients. Furthermore, pre-trained representations often exhibit semantic entanglement, with individual dimensions encoding multiple concepts (*e.g.*, "bird beak" and "bird feet") (Gandelsman et al., 2025; Lim et al., 2025), which can bias models toward dominant but less informative features (*e.g.*, "leaves") while overlooking fine-grained semantics crucial for generation.

To overcome this limitation, we propose a concept-guided tokenizer (ConceptTok), introducing two key innovations. First, instead of aligning with the entire pre-trained features, we project them into a semantic concept space via sparse autoencoders (SAEs) (Gao et al., 2025; Lim et al., 2025) and treat the top-$K$ activated concept indices as alignment signals (Tack et al., 2025), providing sparse, low-dimensional, and disentangled concept supervision. Specifically, we decompose the feature representations of a pre-trained vision–language model (SigLIP (Zhai et al., 2023)) using a TopK SAE (Gao et al., 2025). This supervision encourages the tokenizer to capture fine-grained concept semantics, producing sharper and more consistent gradients by focusing the loss on a few concept indices. More importantly, the tokenizer learns to predict disentangled fine-grained semantic concepts (*e.g.*, "bird beak") rather than imitating entangled feature representations, reducing latent space complexity and promoting compositional transfer (*e.g.*, "bird beak"transferring from jays to crows), and improving generalization in downstream generation.

Second, we enhance the latent space by integrating textual information as an additional input modality. Recognizing that textual descriptions naturally capture higher-level abstractions (Zha et al., 2025; Kim et al., 2025), our tokenizer encoder processes both images and their corresponding text captions to produce a compact latent representation. Unlike prior works that also condition on text at the decoder (de-tokenization) stage (Zha et al., 2025; Kim et al., 2025), which might be biased toward text-to-image generation by the decoder without improving the latent representation itself, our method integrates textual information only into the encoder. This design encourages the latent space to encode complementary semantics, yielding structurally coherent and semantically rich representations that improve generalization in downstream generation tasks.

Our contributions are summarized as follows:

- We propose a concept-guided training objective that advances beyond direct feature alignment. By leveraging activated concept indices in the concept space extracted by an SAE, our method provides fine-grained alignment signals, effectively structuring the latent space around semantic concepts.
- We introduce a novel tokenization framework that, unlike decoder-based textual conditioning methods, integrates textual conditioning only into the encoder. This architectural choice fosters the learning of latent representations that are intrinsically more semantically rich and structurally coherent, providing a stronger foundation for downstream generative models.
- By synergistically combining text-integrated encoding with concept guidance, our approach learns a latent space that excels in both high-fidelity reconstruction and high-level semantic capture, enabling competitive performance in downstream image generation tasks.

## 2 RELATED WORK

**Image Tokenizers** compress high-resolution images into compact tokens within a latent space, which can be either discrete (Van Den Oord et al., 2017; Razavi et al., 2019; Esser et al., 2021; Yu et al., 2022) or continuous (Kingma & Welling, 2014; Li et al., 2024). This transformation enables downstream tasks to operate directly in the compressed latent space, substantially improving efficiency for image generation (Lee et al., 2022; Chang et al., 2022; Rombach et al., 2022) and understanding (Ning et al., 2023). While reconstruction-based training effectively preserves low-level image details (Yu et al., 2024b; Shi et al., 2025), it overlooks semantic structure, limiting generalization to downstream applications (Qu et al., 2025; Xiong et al., 2025b; Lin et al., 2025).

To address this limitation, recent methods integrate explicit semantic guidance into the tokenization process to improve the generalization and utility of the learned latent representations: (1) some approaches align latent features with vision representations extracted from pre-trained models (Qu et al., 2025; Yao et al., 2025; Xiong et al., 2025b), such as CLIP (Radford et al., 2021) or DINOv2 (Oquab et al., 2024); (2) others enforce text–image alignment, encouraging latent features to capture semantics consistent with the images' textual descriptions (Ge et al., 2024; Liang et al., 2024; Wu et al., 2025b); (3) some directly map images into the token space of a frozen large language model (LLM), treating text tokens as the codebook for image representation (Yu et al., 2023; Zhu et al., 2024). However, these strategies align the entire representations, facing the dual challenges of high-dimensional optimization and semantic entanglement. In contrast, our work introduces concept guidance that first projects pre-trained representations into a concept space, providing sparse, low-dimensional, and disentangled alignment signals. Some methods instead directly condition

tokenization on textual input to guide image reconstruction (Zha et al., 2025; Kim et al., 2025). Yet, their performance improvements largely derive from text conditioning applied at the decoder (de-tokenization) stage, which is inherently biased toward text-to-image generation rather than improving the intrinsic quality of the compressed latent representation.

Another family of methods seeks to discover structural or semantic patterns directly from the image. Some approaches partition an image into an adaptive number of arbitrarily shaped regions via segmentation, encoding each region into a token (Wang et al., 2024; Wu et al., 2025a; Chen et al., 2025; Yin et al., 2025). However, the discovered "concepts" typically correspond to concrete pixel regions rather than high-level semantics. Other works represent an image as a set of disentangled visual concept tokens, with each token responding to a distinct visual concept learned solely through reconstruction (Locatello et al., 2020; Yang et al., 2022). However, these methods operate without external semantic supervision and are mainly applied to synthetic datasets (Kim & Mnih, 2018; Gondal et al., 2019), limiting their generalization to complex natural images (Wang et al., 2024).

**Sparse Autoencoders** (SAEs) enforce sparsity in the latent space of an autoencoder by restricting the number of active latent dimensions (Lee et al., 2006). This constraint promotes the learning of disentangled and compact representations that often correspond to coherent semantic concepts (Huben et al., 2024). This capability has led to the broad adoption of SAEs in fields such as natural language processing (Gao et al., 2025; Karvonen et al., 2025) and computer vision (Lim et al., 2025; Zaigrajew et al., 2025), where their ability to produce structured and semantically meaningful concept indices is particularly valuable for model interpretability (Huben et al., 2024), steering model outputs (Lieberum et al., 2024), and enhancing LLM pre-training (Tack et al., 2025).

## 3 PRELIMINARIES

**1D Tokenizer** Our tokenizer is built upon TiTok (Yu et al., 2024b), a vision Transformer (ViT) (Dosovitskiy et al., 2021) based one-dimensional vector-quantized (VQ) (Esser et al., 2021) model. Given an RGB image $\boldsymbol{I} \in \mathbb{R}^{H \times W \times 3}$, where $H$ and $W$ denote the image height and width, respectively, the image is partitioned into non-overlapping patches and linearly projected into patch tokens $\boldsymbol{P} \in \mathbb{R}^{(\frac{H}{f} \times \frac{W}{f}) \times D}$. Here, $f$ is the patch size, $(\frac{H}{f} \times \frac{W}{f})$ is the number of patches, and $D$ is the patch embedding dimension. The patch tokens are concatenated with a set of learnable latent tokens $\boldsymbol{L} \in \mathbb{R}^{N \times D}$, where $N$ denotes the number of learnable latent tokens. The ViT encoder Enc processes this sequence to produce the latent representations:

$$[\_; \boldsymbol{Z}_{1\mathrm{D}}] = \mathrm{Enc}([\boldsymbol{P}; \boldsymbol{L}]), \tag{1}$$

where $[\cdot; \cdot]$ denotes concatenation along the token sequence dimension, the output $\_$ corresponding to the patch tokens $\boldsymbol{P}$ is discarded, and $\boldsymbol{Z}_{1\mathrm{D}} \in \mathbb{R}^{N \times D}$ corresponding to the learnable tokens $\boldsymbol{L}$ serves as the compressed representations used in subsequent steps. Vector quantization (Esser et al., 2021) is applied to $\boldsymbol{Z}_{1\mathrm{D}}$ to obtain discrete latent codes. The ViT decoder Dec takes the quantized tokens $\mathrm{Quant}(\boldsymbol{Z}_{1\mathrm{D}})$ and a new set of learnable patch tokens $\boldsymbol{P}' \in \mathbb{R}^{(\frac{H}{f} \times \frac{W}{f}) \times D}$ to reconstruct the image:

$$[\_; \widehat{\boldsymbol{I}}] = \mathrm{Dec}([\mathrm{Quant}(\boldsymbol{Z}_{1\mathrm{D}}); \boldsymbol{P}']), \tag{2}$$

where $\widehat{\boldsymbol{I}}$ denotes the reconstructed image, while the first $N$ outputs are discarded.

**Sparse Autoencoder** An SAE automatically maps the latent feature representations of a pre-trained model into a semantic concept space; given an image, it extracts a set of associated concept indices within this space. Given the hidden state $\boldsymbol{h} \in \mathbb{R}^{d_h}$ extracted from an image by a large pre-trained vision-language model, the SAE maps $\boldsymbol{h}$ through a linear encoder into high-dimensional activations $\boldsymbol{c} \in \mathbb{R}^{d_c}$, and reconstructs the original input via a linear decoder (Lee et al., 2006). Sparsity constraints are imposed on $\boldsymbol{c}$ to produce compact and interpretable representations, where each active dimension is encouraged to align with a semantically meaningful concept (Huben et al., 2024; Lim et al., 2025). In this work, we adopt a TopK SAE (Makhzani & Frey, 2014; Gao et al., 2025), which enforces sparsity by retaining only the $K$ largest activations.

Formally, the SAE consists of a linear encoder $\mathbf{W}_1 \in \mathbb{R}^{d_h \times d_c}$ and a linear decoder $\mathbf{W}_2 \in \mathbb{R}^{d_c \times d_h}$, with bias terms $\boldsymbol{b}_1 \in \mathbb{R}^{d_c}$ and $\boldsymbol{b}_2 \in \mathbb{R}^{d_h}$. Given a hidden state $\boldsymbol{h} \in \mathbb{R}^{d_h}$, the encoding, sparsification, and reconstruction steps are defined as:

$$\boldsymbol{c}' = \mathbf{W}_1^\top (\boldsymbol{h} - \boldsymbol{b}_2) + \boldsymbol{b}_1, \qquad \boldsymbol{c} = \mathrm{ReLU}\left(\mathrm{TopK}(\boldsymbol{c}')\right), \qquad \hat{\boldsymbol{h}} = \mathbf{W}_2^\top \boldsymbol{c} + \boldsymbol{b}_2, \tag{3}$$

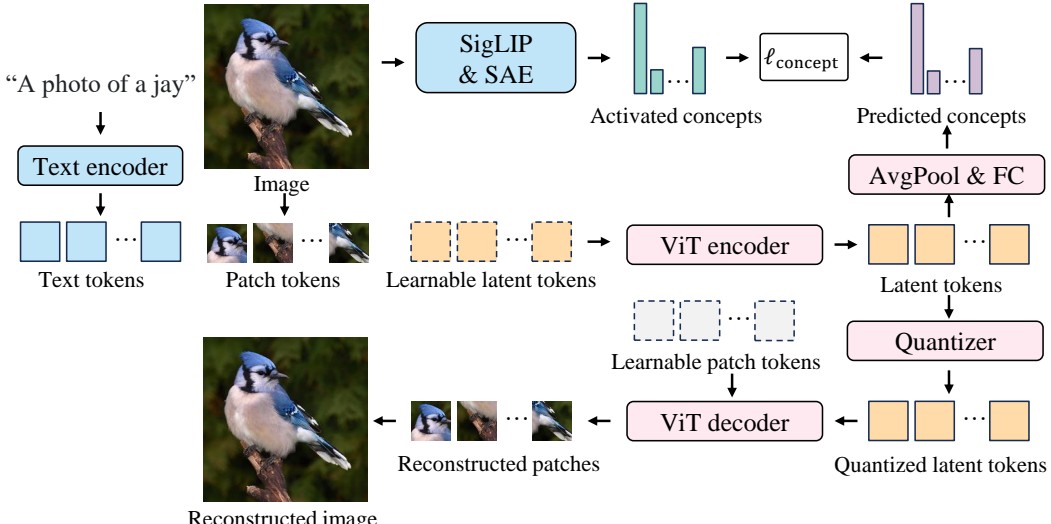

Figure 1: Overview of ConceptTok. The tokenizer encoder (ViT) jointly encodes text tokens, patch tokens, and learnable latent tokens to produce a 1D latent representation. The latent tokens are vector-quantized and, together with learnable patch tokens, passed to the ViT decoder for image reconstruction. During training, SigLIP image features are projected into a sparse concept space via an SAE, and the tokenizer is guided to predict the top-$K$ activated concept indices.

with a training objective that minimizes the reconstruction error, *i.e.*,

$$\ell_{\text{SAE}} = \|\boldsymbol{h} - \hat{\boldsymbol{h}}\|_2^2. \tag{4}$$

Here, $\boldsymbol{c}'$ denotes the pre-activation concept vector, $\text{TopK}(\cdot)$ retains only the $K$ largest activations while setting the rest to zero, followed by $\text{ReLU}$ activation, and $\boldsymbol{c}$ is the resulting sparse concept sets. The reconstruction $\hat{\boldsymbol{h}}$ is obtained by decoding $\boldsymbol{c}$. By enforcing Top-$K$ sparsity, the SAE isolates the most salient dimensions of $\boldsymbol{c}$, each indexing a semantic concept within the image.

## 4 CONCEPT TOKENIZATION

In this section, we present the framework of our ConceptTok, which incorporates text conditioning and concept guidance. The overview of ConceptTok is illustrated in Fig. 1.

### 4.1 TEXT-INTEGRATED TOKENIZER

Most existing methods rely exclusively on image inputs, overlooking accompanying text descriptions as complementary semantic information (Zha et al., 2025). Incorporating such textual cues enriches the latent representation $\boldsymbol{Z}_{1\text{D}}$, thereby improving its effectiveness for downstream tasks like image generation. While some prior approaches incorporate text conditioning (Zha et al., 2025; Kim et al., 2025), their primary focus remains on text-guided image reconstruction at the decoder stage, which is biased toward text-to-image generation rather than enhancing the compressed latent representation. In contrast, our method integrates textual descriptions only into the encoder, explicitly encouraging more semantically meaningful and structurally coherent latent representations.

Given an image and its corresponding text caption, our tokenizer accepts both modalities as input. For the text, we first extract semantic embeddings using a pre-trained CLIP text encoder, followed by a linear projection to align the feature dimension with the patch tokens $\boldsymbol{P}$. This produces text tokens $\boldsymbol{T} \in \mathbb{R}^{T \times D}$, where $T$ denotes the text sequence length. The tokenizer encoder then concatenates the text tokens, patch tokens, and learnable latent tokens $\boldsymbol{L}$, and compresses them into the latent representation:

$$[\_; \_; \boldsymbol{Z}_{1\text{D}}] = \text{Enc}([\boldsymbol{T}; \boldsymbol{P}; \boldsymbol{L}]), \tag{5}$$

where $\boldsymbol{Z}_{1\text{D}}$ associated with $\boldsymbol{L}$ is preserved for subsequent processing, while the outputs corresponding to $\boldsymbol{T}$ and $\boldsymbol{P}$ are discarded. The reconstruction stage follows the same formulation as in Eq. (2).

## 4.2 CONCEPT-GUIDED TRAINING

Prior methods (Yu et al., 2024a; Zha et al., 2025; Kim et al., 2025) typically employ reconstruction-based losses, collectively denoted as $\ell_{\text{VQGAN}}$, which commonly include $\ell_2$ reconstruction loss, perceptual loss (Johnson et al., 2016), adversarial loss with a PatchGAN discriminator (Isola et al., 2017), LeCAM regularization (Tseng et al., 2021), and a VQ codebook loss (Esser et al., 2021). While these objectives facilitate high-fidelity image reconstruction, they predominantly emphasize low-level visual details and often fail to encourage semantically meaningful latent representations (Xiong et al., 2025b; Lin et al., 2025). Recent approaches (Qu et al., 2025; Xiong et al., 2025b)attempt to address this by aligning latent features with those from pre-trained models.

Unfortunately, directly aligning with entire pre-trained feature representations still suffers from both optimization and generalization challenges. First, features from pre-trained models are high-dimensional (*e.g.*, 768 for CLIP/B or DINOv2/B), making such alignment a high-dimensional regression problem where the curse of dimensionality flattens distance metrics and weakens optimization gradients. Second, holistic feature alignment operates on entangled embeddings, where each dimension encodes mixed semantic concepts (*e.g.*, "bird beak" and "bird feet") (Lim et al., 2025). Consequently, gradients are spread thinly across many entangled dimensions, likely biasing the model towards dominant but less informative features (*e.g.*, "leaves") while diluting the fine-grained signals most critical for generation.

To address these limitations, we introduce a concept-guided training objective (Tack et al., 2025) that encourages the latent $\boldsymbol{Z}_{\text{1D}}$ to capture fine-grained concept semantics. Rather than aligning with the entire pre-trained features, we project pre-trained features into a semantic concept space via a TopK SAE (Gao et al., 2025) and treat the top-$K$ (*e.g.*, $K = 128$) activated concept indices as alignment signals. Such sparse, low-dimensional, and disentangled supervision offers several merits: by focusing the loss on a few activated concepts, the optimization becomes more stable and efficient with sharper and more consistent gradients; more importantly, the tokenizer learns to predict disentangled semantic concepts (*e.g.*, "bird beak") rather than imitate entangled embeddings, thereby reducing latent space complexity, promoting compositional transfer (*e.g.*, "bird beak" transferring from jays to crows) and facilitating generalization in downstream generation.

Formally, we utilize a pre-trained vision-language model (*e.g.*, SigLIP (Zhai et al., 2023)) that captures and aligns rich visual-textual semantics to discover semantic concepts. We train a TopK SAE on features from the last layer of its vision encoder using the LLaVA-NeXT dataset (Liu et al., 2024) that provides high-quality aligned image-text pairs with rich semantic correspondence. The SAE produces a concept space from which we extract the indices of the top-$K$ activated concepts. Let $\mathcal{I} = \{i_1, i_2, \ldots, i_K\}$ denote the set of indices corresponding to the top-$K$ entries in the SAE's activations $\boldsymbol{c}$. To inject this semantic information into our tokenizer, we first average-pool the latent sequence $\boldsymbol{Z}_{\text{1D}}$ and then apply a fully-connected (FC) layer $\phi$ to obtain a concept prediction vector:

$$\boldsymbol{z} = \phi\left(\text{AvgPool}(\boldsymbol{Z}_{\text{1D}})\right). \tag{6}$$

The concept loss is then computed as:

$$\ell_{\text{concept}} = \frac{1}{K} \sum_{i \in \mathcal{I}} \text{CE}(\boldsymbol{z}, i) \tag{7}$$

where CE denotes the cross-entropy loss. This objective directly encourages the model to correctly identify the activated indices in the concept space, thereby enhancing the semantic structure of the latent space. The complete training objective combines the concept loss with traditional reconstruction losses:

$$\ell_{\text{total}} = \ell_{\text{VQGAN}} + \lambda \cdot \ell_{\text{concept}} \tag{8}$$

where $\lambda$ is a weighting hyperparameter. This approach ensures that the tokenizer learns to reconstruct input images while simultaneously capturing high-level semantic concepts, resulting in more meaningful latent representations for downstream vision tasks.

## 5 EXPERIMENTS

In this section, we first describe the experimental setups and then present comparison results.

## 5.1 EXPERIMENT SETUPS

**Tokenization Models**   We implement our tokenizer using ViT-based architectures, specifically employing ViT/S, ViT/B, and ViT/L as encoder and decoder components. For instance, a model designated as "B-L" uses a ViT/B encoder and a ViT/L decoder. Following the setup in (Kim et al., 2025), input images are of resolution $256 \times 256$ and divided into patches of size $16 \times 16$, resulting in 256 patch tokens. To accommodate varying compression ratios, we vary the number of latent tokens as $N \in \{32, 64, 128\}$. For the vector-quantized model, a codebook of $8,192$ entries is used, each with 64 channels, as in (Kim et al., 2025).

**Class-conditional Image Generative Models**   Following (Xiong et al., 2025b), we evaluate our tokenizer's applicability to the class-conditional image generation task using two variants of Llama-Gen (Sun et al., 2024): LlamaGen-B and LlamaGen-XL. Class conditioning is implemented through learnable embeddings (Esser et al., 2021), which act as prefilling class tokens indicating the specific ImageNet class. Beginning with the class token, the generative model autoregressively predicts a sequence of latent tokens, whose length matches the number of latent tokens of the tokenizer. These predicted tokens are then passed through the pre-trained tokenizer decoder to get the final image.

**Training Setups**   We train a TopK SAE (Gao et al., 2025) to map the image representations from a pre-trained SigLIP-B/16 vision encoder (Zhai et al., 2023) into a semantic concept space. The SAE is trained on final-layer features from the SigLIP vision encoder, which are aligned with text embeddings and thus inherently semantic. SAE training is performed on the LLaVA-NeXT dataset (Liu et al., 2024), with the concept space dimension set to $d_c = 24{,}576$ and sparsity parameter $K = 128$. We adopt a learning rate of $4 \times 10^{-4}$ with 500 warm-up steps, following (Lim et al., 2025).

For our main tokenization framework, both tokenization and generative models are trained on ImageNet training set (Russakovsky et al., 2015) at $256 \times 256$ resolution with standard data augmentations including random cropping and horizontal flipping, following previous works (Yu et al., 2024b). Since ImageNet lacks captions, we construct class-descriptive text prompts using the template "*A photo of a {class name}*" (Kim et al., 2025). Tokenization models are trained for 200 epochs with a maximum learning rate of $10^{-4}$ and a cosine decay schedule (Loshchilov & Hutter, 2017), with a default trade-off $\lambda = 0.1$ to balance reconstruction and concept guidance objectives. For downstream evaluation, generative models are trained for 300 epochs using the WSD scheduler (Hägele et al., 2024) with a base learning rate of $10^{-4}$, decay ratio of 0.2, and 1-epoch warm-up, consistent with (Xiong et al., 2025b).

**Evaluation Metrics**   We evaluate reconstruction quality using reconstruction Fréchet Inception Distance (rFID) (Heusel et al., 2017) and reconstruction Inception Score (rIS) (Salimans et al., 2016) on ImageNet validation set (Russakovsky et al., 2015) at $256 \times 256$ resolution. To assess downstream image generation performance, we train the class-conditional autoregressive (AR) image generative models (*i.e.*, LlamaGen (Sun et al., 2024)) using the learned tokenizer on ImageNet and report generation Fréchet Inception Distance (gFID) and generation Inception Score (gIS), following established evaluation protocols in ADM (Dhariwal & Nichol, 2021).

## 5.2 MAIN RESULTS

We evaluate our ConceptTok against other state-of-the-art tokenizers on ImageNet $256 \times 256$ reconstruction and generation benchmark, as shown in Tab. 1. In terms of reconstruction, ConceptTok achieves competitive reconstruction fidelity, with ConceptTok-B-L-64 attaining a high rIS of 325.8 and ConceptTok-B-L-128 attaining a comparable rFID of LlamaGenTok (Sun et al., 2024).

Crucially, the semantic structure inherent in our tokenizer enables strong generalization to downstream image generation tasks, achieving highly competitive performance and efficiency with a smaller number of latent tokens. When paired with the autoregressive LlamaGen-XL generator (775M), ConceptTok-B-L-128 achieves a highly competitive gFID of 2.37 and gIS of 248.7. This performance is on par with significantly larger autoregressive models like Open-MAGVIT2-XL (1.5B, 2.33 gFID) and IBQ-XXL (2.1B, 2.05 gFID), despite our generator being substantially smaller. Moreover, by operating on sequences of only 128 or 64 latent tokens—half to a quarter the length of the 256 tokens used by other discrete tokenizers—our method achieves a superior trade-off

Table 1: Main results on ImageNet $256 \times 256$. Reconstruction performance of ConceptTok is evaluated on ImageNet validation set, while generation performance follows the evaluation protocols in ADM (Dhariwal & Nichol, 2021) for fair comparison in class-conditional generation. "Type" specifies the generative model family, where "Diff.", "AR" and "Mask." correspond to diffusion models, autoregressive models, and masked Transformer models, respectively. ‡: training includes additional data beyond ImageNet. ⋆: class-conditional generation without classifier-free guidance.

| Tokenizer | Param. | #Tokens | rFID↓ | rIS↑ | Generator | Param. | Type | gFID↓ | gIS↑ |
|---|---|---|---|---|---|---|---|---|---|
| **Continuous tokens** | | | | | | | | | |
| VAE (Rombach et al., 2022) | 55M | 4096 | 0.27 | – | LDM-4 (Rombach et al., 2022) | 400M | Diff. | 3.60 | – |
| SD-VAE (Ma et al., 2024) | 84M | 1024 | 0.62 | – | SiT-XL/2 (Ma et al., 2024) | 675M | Diff. | 2.06 | – |
| VA-VAE (Yao et al., 2025) | 70M | 256 | 0.28 | 205.6 | LightningDiT (Yao et al., 2025) | 675M | Diff. | 1.35 | 295.3 |
| VAE (Li et al., 2024) | 66M | 256 | 0.53 | – | MAR-H (Li et al., 2024) | 943M | AR+Diff. | 1.55 | 303.7 |
| **Discrete tokens** | | | | | | | | | |
| B-AE-d32 (Wang et al., 2023) | 66M | 256 | 1.69 | – | BiGR-XXL-d32 (Hao et al., 2025) | 1.5B | AR+Diff | 2.36 | 277.2 |
| VQGAN (Chang et al., 2022) | 66M | 256 | 2.28 | – | MaskGIT (Chang et al., 2022) | 227M | Mask. | 6.18⋆ | – |
| TiTok-B (Yu et al., 2024b) | 202M | 64 | 1.70 | – | MaskGIT-ViT (Chang et al., 2022) | 177M | Mask. | 2.48 | 214.7 |
| TiTok-L (Yu et al., 2024b) | 641M | 32 | 2.21 | – | MaskGIT-ViT (Chang et al., 2022) | 177M | Mask. | 2.77 | 199.8 |
| VAR-Tok (Tian et al., 2024) | 109M | 680 | 1.00‡ | – | VAR-d24 (Tian et al., 2024) | 1.0B | VAR | 2.09 | 312.9 |
| VAR-Tok (Tian et al., 2024) | 109M | 680 | 1.00‡ | – | VAR-d30 (Tian et al., 2024) | 2.0B | VAR | 1.92 | 323.1 |
| ImageFolder (Li et al., 2025) | 176M | 286 | 0.80‡ | – | ImageFolder-VAR (Li et al., 2025) | 362M | VAR | 2.60 | 295.0 |
| VQGAN (Esser et al., 2021) | 23M | 256 | 4.98 | – | Taming-Transformer(Esser et al., 2021) | 1.4B | AR | 15.8⋆ | 78.3⋆ |
| ViT-VQGAN (Yu et al., 2022) | 64M | 1024 | 1.28 | 192.3 | VIM-Large (Yu et al., 2022) | 1.7B | AR | 4.17⋆ | 175.1⋆ |
| RQ-VAE (Lee et al., 2022) | 66M | 256 | 3.20 | – | RQ-Transformer (Lee et al., 2022) | 3.8B | AR | 7.55⋆ | 134.0⋆ |
| Open-MAGVIT2 (Luo et al., 2024) | 133M | 256 | 1.17 | – | Open-MAGVIT2-B (Luo et al., 2024) | 343M | AR | 3.08 | 258.3 |
| Open-MAGVIT2 (Luo et al., 2024) | 133M | 256 | 1.17 | – | Open-MAGVIT2-XL (Luo et al., 2024) | 1.5B | AR | 2.33 | 271.8 |
| IBQ (Shi et al., 2025) | 128M | 256 | 1.37 | – | IBQ-B (Shi et al., 2025) | 342M | AR | 2.88 | 254.7 |
| IBQ (Shi et al., 2025) | 128M | 256 | 1.37 | – | IBQ-XXL (Shi et al., 2025) | 2.1B | AR | 2.05 | 286.7 |
| LlamaGenTok (Sun et al., 2024) | 72M | 256 | 2.19 | – | LlamaGen-B (Sun et al., 2024) | 111M | AR | 5.46 | 193.6 |
| LlamaGenTok (Sun et al., 2024) | 72M | 256 | 2.19 | – | LlamaGen-XL (Sun et al., 2024) | 775M | AR | 3.39 | 227.1 |
| GiGaTok-B-L (Xiong et al., 2025b) | 622M | 256 | 0.81 | – | LlamaGen-B (Sun et al., 2024) | 111M | AR | 3.26 | 221.0 |
| GiGaTok-XL-XXL (Xiong et al., 2025b) | 2.9B | 256 | 0.79 | – | LlamaGen-B (Sun et al., 2024) | 111M | AR | 3.15 | 224.3 |
| ConceptTok-B-B | 316M | 64 | 3.84 | 276.0 | LlamaGen-B (Sun et al., 2024) | 111M | AR | 4.13 | 245.8 |
| ConceptTok-B-L | 533M | 64 | 4.52 | 325.8 | LlamaGen-B (Sun et al., 2024) | 111M | AR | 3.28⋆ | 254.2⋆ |
| ConceptTok-B-L | 533M | 128 | 2.38 | 285.8 | LlamaGen-B (Sun et al., 2024) | 111M | AR | 2.97 | 232.4 |
| ConceptTok-B-L | 533M | 128 | 2.38 | 285.8 | LlamaGen-XL (Sun et al., 2024) | 775M | AR | 2.37⋆ | 248.7⋆ |

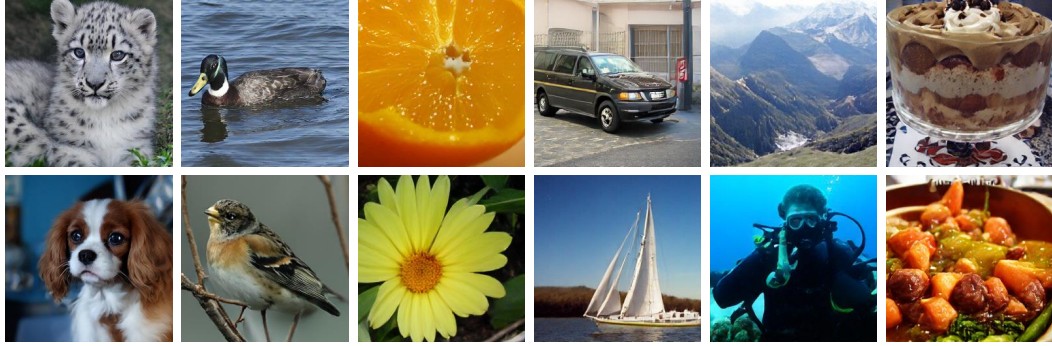

Figure 2: Class-conditional generation results from ConceptTok-B-L-128 using the LlamaGen-XL framework, producing high-fidelity images with fine-grained concept details.

between generation quality and inference efficiency. When paired with the same LlamaGen-B generator, ConceptTok-B-L-128 (533M, 2.97 gFID) not only outperforms the GiGaTok-B-L (622M, 3.26 gFID) but also surpasses the much larger GiGaTok-XL-XXL (2.9B, 3.15 gFID). This demonstrates that our concept guidance yields more semantically structured latent representations than holistic feature alignment, leading to superior generalization in downstream image generation tasks.

As a qualitative demonstration of this capability, Fig. 2 presents class-conditional generative images from our Ours-B-L-128 tokenizer paired with the LlamaGen-XL generator. The samples exhibit semantically rich structures and finely rendered concept details, visually corroborating the high quantitative scores achieved by our method.

## 5.3 CONCEPT ALIGNMENT ANALYSIS

**Quantitative Results** To quantitatively evaluate the semantic alignment between our tokenizer and the SAE-derived concept indices by computing the F1-score for each ImageNet validation im-

Table 2: Concept alignment results on ImageNet validation set, measured by F1-score between tokenizer predictions and SAE-derived concept indices, demonstrating a high semantic overlap.

| Tokenizer | F1-score |
|---|---|
| ConceptTok-B-B-64 | 0.601 |
| ConceptTok-B-L-64 | 0.618 |
| ConceptTok-B-L-128 | 0.623 |

Table 3: Ablation results of the number of latent tokens $N$. Increasing $N$ improves both reconstruction and generation performance.

| Tokenizer | Reconstruction | | Generation | |
|---|---|---|---|---|
| | rFID↓ | rIS↑ | gFID↓ | gIS↑ |
| B-L-32 | 7.52 | 332.1 | 6.19 | 297.7 |
| B-L-64 | 4.52 | 325.8 | 3.28 | 254.2 |
| B-L-128 | 2.38 | 285.8 | 2.97 | 232.4 |

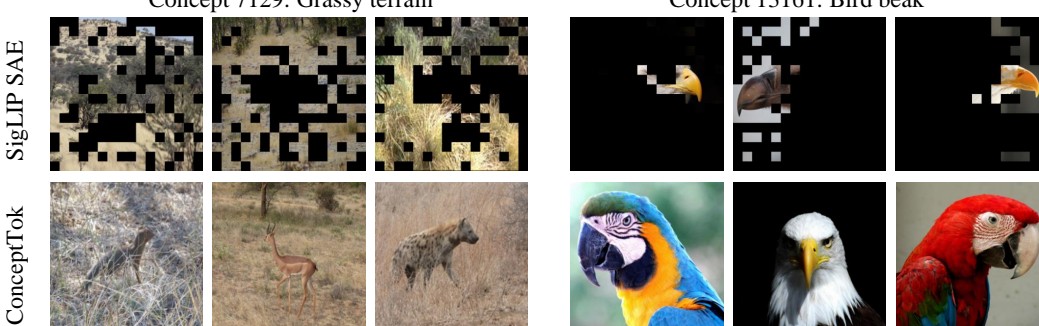

Figure 3: Concept alignment visualization. Top row: image patches that most strongly activate a specific concept index in the SigLIP SAE's concept space. Bottom row: images where our tokenizer's latent representation yields the highest scores for the corresponding concept index. The retrieved images consistently contain the corresponding semantics across different contexts, demonstrating strong fine-grained concept alignment.

age. This metric measures the overlap between the top-$K$ concept indices identified by the pretrained SigLIP SAE and those predicted by our tokenizer. As shown in Tab. 2, increasing both model size and latent token count improves concept alignment, demonstrating that larger models with expanded latent tokens better capture SAE-derived semantic concepts. This enhanced alignment supports improved performance in downstream generative tasks shown in Tab. 1.

**Qualitative Results** We provide qualitative visualizations on ImageNet validation set to illustrate concept alignment. For each concept index identified by the SigLIP SAE, we retrieve images with the highest activation values and highlight the specific patches whose SigLIP representations activate the concept index while masking other regions (Lim et al., 2025). As shown in Fig. 3, the top row presents patch-level visualizations for example concept indices (*e.g.*, those corresponding to "grassy terrain" and "bird beak"), where the unmasked regions precisely localize the semantic features associated with each concept index. The bottom row displays images for which our tokenizer's latent representation $Z_{1D}$ produces the highest scores for these same target concept indices. Notably, the retrieved images consistently contain the corresponding semantics (*e.g.*, grassy terrain or bird beak) regardless of contextual variations, such as antelopes and leopards in grassy terrain. This consistency across contexts demonstrates that our tokenizer achieves fine-grained concept alignment with the pre-trained model's representations, as evidenced by the strong correspondence between SAE-activated patches and tokenizer-retrieved images.

**Latent Space Analysis** We further analyze the structure of the learned latent space using t-SNE visualizations (Maaten & Hinton, 2008), as shown in Fig. 4, with example images for SAE-derived concept indices provided in Fig. 5 in Appendix. Fig. 4 shows that our method produces latent representations that form distinct clusters corresponding to semantic concepts, indicating successful alignment of the representation space around a semantic concept space. This structured latent space reduces complexity and facilitates more effective training of downstream generative models. In contrast, GigaTok's representations are poorly separated, lacking distinct clusters for individual concepts.

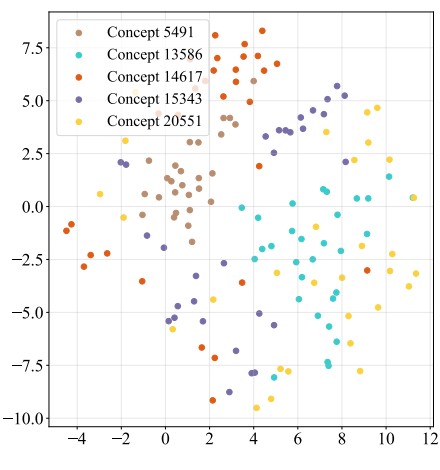

(a) t-SNE of GigaTok latent embeddings.

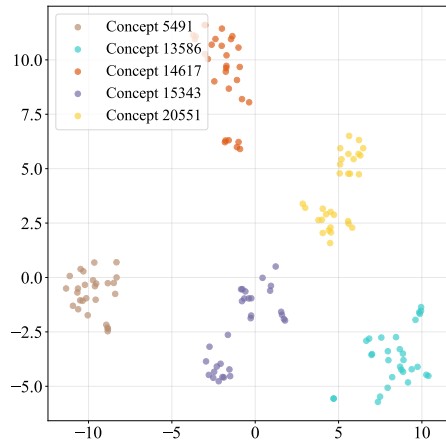

(b) t-SNE of ConceptTok latent embeddings.

Figure 4: Comparison of latent space structure. (a) GigaTok embeddings are semantically entangled. (b) ConceptTok embeddings form discernible semantic clusters, demonstrating more structured representations with reduced complexity that benefit downstream image generation.

## 5.4 ABLATION STUDIES

**Key Components**   We perform an ablation study to assess the contribution of each component in ConceptTok. Quantitative results in Tab. 4 demonstrate a clear performance trend: the baseline tokenizer yields an rFID of 7.95 and a gFID of 7.31. The introduction of text conditioning leads to a significant improvement, reducing rFID to 4.66 and gFID to 5.31, underscoring the benefit of integrating textual semantics. The inclusion of concept guidance further elevates performance, achieving the best scores of rFID 3.84 and gFID 4.13. This progressive enhancement confirms that both text conditioning and concept guidance are crucial for learning semantically structured representations that effectively generalize to image generation tasks.

Table 4: Ablation study of ConceptTok on B-B-64. Text conditioning improves performance over the baseline, while concept guidance yields further gains, validating each component's contribution.

| Componets | rFID↓ | rIS↑ | gFID↓ | gIS↑ |
|---|---|---|---|---|
| Baseline | 7.95 | 96.5 | 7.31 | 170.1 |
| + Text conditioning | 4.66 | 284.8 | 5.31 | 250.3 |
| + Text conditioning + Concept guidance | 3.84 | 276.0 | 4.13 | 245.8 |

**Tokenizer Variants**   We analyze the impact of the number of latent tokens on tokenization performance. As shown in Tab 3, increasing from 32 to 128 tokens reduces rFID from 7.52 to 2.38 and gFID from 6.19 to 2.97, demonstrating that more tokens capture richer information. We also examine model scale effects by comparing architectures with 64 tokens, as shown in Tab. 6 in Appendix. Larger models consistently improve downstream generation quality, highlighting the benefit of increased capacity for learning structured latent representations.

## 6 CONCLUSION AND DISCUSSION

We present ConceptTok, a novel tokenization framework that integrates text conditioning and concept guidance to improve the semantic structure of the latent space. The resulting representations demonstrate strong concept alignment and generalize effectively to downstream image generation tasks. Future research directions include scaling ConceptTok training to broader multimodal datasets and applying it to various tasks such as text-to-image generation and image understanding.

ETHICS STATEMENT

This work utilizes publicly available resources, including ImageNet, LLaVA-NeXT, and the pre-trained SigLIP model, all of which are widely adopted within the research community. We acknowledge that these resources may contain inherent knowledge that may be reflected in our tokenizer's outputs. We also recognize the risks associated with the misuse of image reconstruction and generation technologies, particularly in sensitive contexts. Consequently, we emphasize that deploying such technologies requires careful ethical consideration.

REPRODUCIBILITY STATEMENT

Details of our experimental setup are provided in Section 5.1 and Appendix B. All resources utilized in this work, including datasets and pre-trained models such as SigLIP, are publicly accessible. Our implementation code will be made publicly available on GitHub upon acceptance of the paper.

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

# A  LLM USAGE STATEMENT

LLMs were used to assist in polishing the writing, including improving grammatical correctness, sentence fluency, and overall academic style. All technical content, experimental results, and intellectual contributions remain entirely our own. The LLM was used solely as a writing enhancement tool and did not contribute to the scientific reasoning or methodological development of this work.

# B  MORE IMPLEMENTATION DETAILS

## B.1  SAE TRAINING DETAILS

We use SigLIP-B/16 (Zhai et al., 2023) with an input resolution of $256 \times 256$, taking the final layer of the vision encoder, which has a feature dimension of 768. For the TopK SAE, we set the number of concepts to $d_c = 24{,}576$ and the sparsity parameter to $K = 128$. The SAE is trained with a learning rate of $4 \times 10^{-4}$, a constant warm-up scheduling with 500 warm-up steps (Lim et al., 2025). Decoder biases are initialized with the geometric median. Training is performed for 13,640 iterations with a batch size of 192 on the LLaVA-NeXT dataset (Liu et al., 2024), using ghost gradients for optimization (Lim et al., 2025).

## B.2  TOKENIZER TRAINING DETAILS

We provide detailed training hyperparameters for our tokenizers in Tab. 5

Table 5: Training hyperparameters.

| Hyperparameter | Value |
|---|---|
| $\ell_2$ loss weight | 1.0 |
| Quantizer loss weight | 1.0 |
| Concept loss weight | 0.1 |
| Adversarial loss weight | 0.1 |
| Discriminator starting epoch | 80 |
| Perceptual loss weight | 1.1 |
| Perceptual loss models | LPIPS VGG (Zhang et al., 2018) ConvNeXt Small (Liu et al., 2022) |
| LeCAM weight | 0.001 |
| Learning rate | $10^{-4}$ |
| Optimizer | AdamW (Loshchilov & Hutter, 2019) ($\beta_1 = 0.9, \beta_2 = 0.999$) |
| Learning rate schedule | Cosine learning decay (Loshchilov & Hutter, 2017) |
| Weight decay | $10^{-4}$ |
| Training epochs | 200 |
| Batch size | 256 / 512 |

## B.3  IMAGE GENERATION DETAILS

**Class-free Guidance**  The rFID of generative models can be significantly influenced by classifier-free guidance (CFG) (Ho & Salimans, 2021; Sun et al., 2024). To be consistent with previous work (Xiong et al., 2025b), we perform a grid search for the optimal CFG scale within the range 1.0 to 3.0 with step size 0.25. Specifically, we follow the approach in (Xiong et al., 2025b) where models generate the first 18% of tokens without guidance (*i.e.*, CFG scale $= 1.0$) to encourage image diversity, after which CFG is applied to the remaining tokens to enhance visual quality.

# C  MORE RESULTS

We also examine model scale effects by comparing architectures with 64 tokens, as shown in Tab. 6.

Table 6: Impact of model scale on tokenization performance.

| Tokenizer | Param. | rFID↓ | rIS↑ | gFID↓ | gIS↑ |
|---|---|---|---|---|---|
| S-S-64 | 189M | 6.68 | 192.0 | 7.13 | 241.2 |
| B-B-64 | 316M | 3.84 | 276.0 | 4.13 | 245.8 |
| B-L-64 | 533M | 4.52 | 325.8 | 3.28 | 254.2 |

## D  CONCEPT VISUALIZATION

To interpret the learned representations, we visualize concepts derived from the SAE by identifying images that yield the highest activation for each concept index, as shown in Fig. 5. Following (Zhang et al., 2025), we then use a multimodal large language model (*e.g.*, Qwen2.5-VL (Bai et al., 2025)) to generate descriptive names for these concepts based on their corresponding image sets. Finally, two authors independently verify the appropriateness of the proposed concept names to ensure semantic consistency.

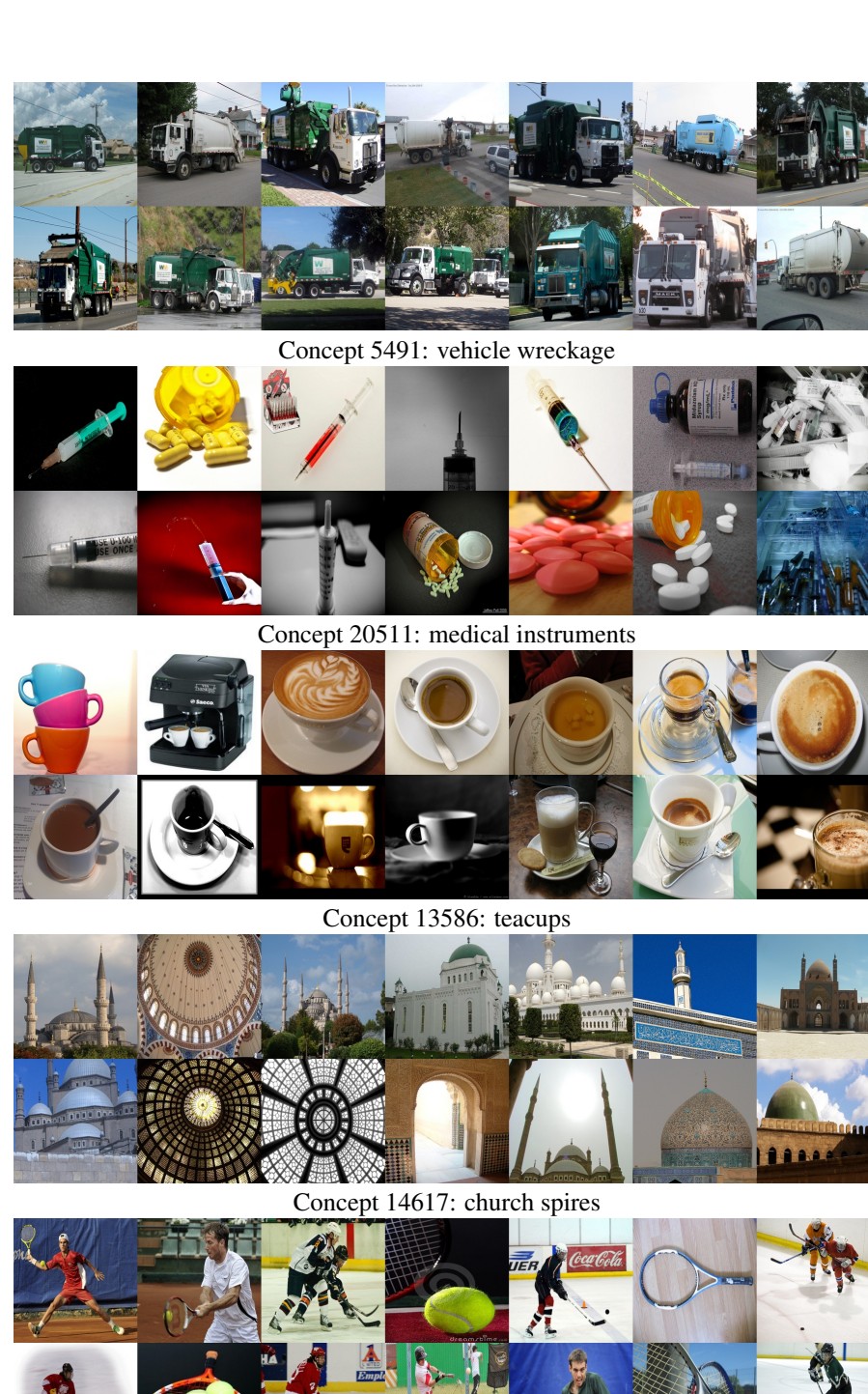

Concept 5491: vehicle wreckage

Concept 20511: medical instruments

Concept 13586: teacups

Concept 14617: church spires

Concept 15343: sports equipment handles

Figure 5: Example images for selected SAE-derived concept indices.

