# OpenReview forum: "Concept-Guided Tokenization: Closing the Gap Between Reconstruction and Generation"
_ICLR.cc/2026/Conference — Submitted to ICLR 2026_

### Official Review · Reviewer_Kkz9 · 2025-10-25

**Soundness:** 2
**Presentation:** 3
**Contribution:** 2
**Rating:** 4
**Confidence:** 4

**Summary:**

The paper proposes ConceptTok, a concept-guided image tokenizer that integrates text conditioning and concept-level supervision via a Sparse Autoencoder (SAE). It leverages SigLIP features to construct a sparse concept space, using the top-K activated concept indices to guide the tokenizer toward learning disentangled and semantically meaningful latent representations. This design aims to enhance both reconstruction fidelity and image generation quality.

**Strengths:**

1.	Introduces a clear framework for incorporating semantic concept guidance into tokenizer training.
2.	Demonstrates strong quantitative and qualitative improvements in generation quality with fewer latent tokens.

**Weaknesses:**

1.	The paper enforces strong semantic alignment with a fixed SigLIP+SAE concept space, which may over-constrain the latent representation. Such semantic over-alignment could hurt generalization to unseen or open-domain concepts, limiting the tokenizer’s flexibility beyond ImageNet-like settings.
2.	The paper assumes a causal link between the tokenizer’s concept-understanding ability and improved image generation quality, which is not well justified. A tokenizer can still enable high-quality generation even without explicit concept-level understanding, as long as the autoregressive model effectively models token composition. Moreover, the paper does not provide evidence that the SAE can effectively capture fine-grained semantic information that directly benefits image generation.
3.	The performance improvement is more likely due to the strong supervision from SigLIP features, which provides a powerful alignment signal. The concept loss primarily serves as a mechanism to better align the tokenizer with SigLIP features, rather than demonstrating that the “concepts” themselves are crucial for reconstruction or generation quality.

**Questions:**

Q1. Could the authors clarify why learning semantic or concept-level information within the tokenizer necessarily leads to better image generation quality? What is the causal relationship between semantic alignment and generative performance?

Q2. Given that the tokenizer and codebook are trained to align with a fixed set of SigLIP+SAE concepts, how would this approach generalize to cross-domain or text-to-image settings? If the target concepts fall outside the predefined SAE concept space, would the tokenizer still function effectively?

---

### Official Review · Reviewer_NKTd · 2025-10-31

**Soundness:** 3
**Presentation:** 3
**Contribution:** 3
**Rating:** 6
**Confidence:** 3

**Summary:**

This paper introduces ConceptTok, a tokenization framework for images that aims to bridge the historical gap between high-fidelity reconstruction and semantically driven image generation. The method integrates a text-conditioned encoder, allowing the tokenizer to ingest both visual and linguistic cues, and introduces concept-guided supervision by leveraging sparse autoencoders trained to decompose vision-language model features into a semantic concept space. The framework enables the learned latent tokens to align with fine-grained, disentangled high-level concepts. Results are presented on ImageNet 256×256 benchmarks for both reconstruction and downstream generation, supported by quantitative and qualitative analyses.

**Strengths:**

1. The approach pairs a text-integrated visual tokenizer with concept-based supervision. By incorporating a TopK sparse autoencoder projection of pre-trained SigLIP features, the method aims to yield disentangled, semantic, and interpretable guidance, which is a step forward over direct feature alignment.
2. The separation of text conditioning into the encoder is clearly justified and evaluated, showing the impact on semantic structure in the latent space.
3. The method is evaluated both for reconstruction fidelity and for downstream compositional generation, with consistent benchmarks. The results show competitive or superior trade-offs in generation/reconstruction with fewer latent tokens.

**Weaknesses:**

- The text encoder for constructing T uses CLIP, but there are no experiments or ablations to examine robustness if text encoders or textual styles are varied.
- While the paper leverages SAE-derived concept indices, there is insufficient analysis of what happens when the SAE fails to capture relevant concepts or when concept activations are ambiguous. The reliance on a fixed K=128 top concept may not be optimal across all images with varying semantic complexity.
- The study is limited to ImageNet, evaluations on caption-rich datasets would strengthen claims about semantic generalization.

**Questions:**

+ The paper uses λ=0.1 for concept loss weighting. How was this value chosen? What is the sensitivity to this hyperparameter? Is there a trade-off curve between reconstruction and semantic alignment?
+ Would fine-tuning the SAE jointly with the tokenizer, instead of freezing it, further improve alignment?
+ How does ConceptTok perform with richer text inputs, e.g., multi-sentence descriptions or compositional prompts?

---

### Official Review · Reviewer_BG28 · 2025-10-31

**Soundness:** 3
**Presentation:** 3
**Contribution:** 2
**Rating:** 4
**Confidence:** 2

**Summary:**

This paper proposes a method called ConceptTok to improve the quality of tokens in image tokenizers. The main idea is that, in addition to the reconstruction task for tokenization, semantic information can be used as an additional loss term. The semantic information is derived via a parallel processing branch featuring sparse autoencoders. The paper reports experimental results on ImageNet 256x256 and compares against a wide range of tokenizers for reconstruction and generation quality.

Overall, the paper's motivation for using semantic information to guide the image tokenizer is intuitive, but the current results do not convince readers that the proposed method is the most effective solution.

**Strengths:**

- The idea of using fine-grained visual concepts to guide the learning of tokenizers is intuitive. While one could argue that the reconstruction task itself is implicitly relying on the semantic understanding of the image, having explicit semantic guidance should certainly help improve the quality of the visual tokens.

- The structure and the writing of the paper are easy to follow.

**Weaknesses:**

- My main concern is that the performance of the proposed tokenization does not show meaningful improvement over alternatives. Table 1 shows a comprehensive view of the comparison of the proposed ConceptTok vs. other tokenizers. The result looks pretty mixed: ConceptTok has a higher rIS score (although most of the methods have blank performance in this column), but it takes more parameters than others and it has a significantly worse rFID than some tokenizers (such as VAE). It is unclear if the proposed tokenization provides a significant benefit in real applications.

- There is a lack of in-depth analysis of the claimed benefits of the ConceptTok. For example, one of the main contributions mentioned in the introduction is that the proposed method 'provides fine-grained alignment signals'. There is no convincing quantitative result on this: the closest is in Table 2 where the semantic classification results are shown. This is not enough because that's the task the tokenizer is trained on. The authors need a more comprehensive analysis to better show the significance of 'fine-grained alignment'.

- The experiment is only done on one dataset ImageNet 256x256.

**Questions:**

1. Can the authors discuss more about the mixed performance of the proposed method in Table 1? Also, those tokenizers and image generators have different parameter settings and backbones, can the authors discuss what would be a fair comparison to assess the effectiveness of the tokenizers?

2. The authors need to expand their analysis on the concept alignment.

3. The authors should present results on more datasets.

---

### Official Review · Reviewer_DqCj · 2025-11-01

**Soundness:** 3
**Presentation:** 4
**Contribution:** 3
**Rating:** 4
**Confidence:** 3

**Summary:**

This paper proposes ConceptTok, a concept-guided image tokenization framework, aiming to solve the issue that existing tokenizers excel at preserving low-level visual details via reconstruction training but lack explicit semantic guidance, hindering downstream image generation. It introduces two key innovations: a text-integrated encoder fusing images and textual descriptions in the encoder and a concept-guided objective using Top-K SAE to decompose SigLIP features into disentangled semantic concepts. Trained on ImageNet and LLaVA-NeXT, ConceptTok balances reconstruction and generation, matching large models with fewer latent tokens.

**Strengths:**

1. The motivation is clear and the writing is easy to follow.
2. The method is simple yet effective.
3. Backgrounds on 1D tokenizer and SAE are introduced.

**Weaknesses:**

1. All experiments in the paper adopt the combination of SigLIP and SAE, yet the paper did not explain why SigLIP must be used specifically. There is no verification to prove that the improved performance originates from SAE rather than SigLIP itself. To address this ambiguity, ablation experiments should be conducted by replacing SigLIP with other pre-trained models (such as CLIP, DINOv2).

2. It can be observed from Tab. 1 that ConceptTok has a much larger parameter count than LlamaGenTok. Despite the larger parameter size, ConceptTok’s rFID is not superior to LlamaGenTok. The paper does not provide comparative experiments that align the two models in terms of parameter count or token count, making it difficult to fairly evaluate whether ConceptTok’s design itself brings advantages.

3. The paper additionally introduces text input into the tokenizer, but only conducts experiments on class-to-image generation. In this task, the pre-defined class learnable embeddings essentially serve a similar role to text input. Furthermore, it does not perform experiments on text-to-image (t2i) datasets, leaving unclear whether the text-integrated encoder can truly leverage more flexible textual semantics to improve generation performance in scenarios beyond class conditioning.

**Questions:**

1. Is the TopK SAE used in this paper a novel method proposed by the authors or an existing technique?

2. In Tab. 1, which methods also utilize semantic guidance? What specific types of guidance are used? These details should be clarified.

3. Tab. 3 lacks results for the "Baseline + concept guidance" setting.

---

### Meta-Review · Area_Chair_jgid · 2025-12-30

**Summary:**

This paper proposes a 1D image tokenizer that aligns the token representation to semantic concepts in the image. Experimental results are presented on ImageNet 256x256 with mixed setups for tokenizer and generator models. All reviewers raised concerns around the insufficiency of empirical evidence, both in terms of controlled comparison against earlier works and in terms of demonstrating the effectiveness of the added semantic concepts for improving generation quality. No rebuttal was posted, and no discussion occurred.

**Reviewer Concerns:**

N/A, as no rebuttal was submitted

**Reviewer Scores:**

N/A, as no rebuttal was submitted

---

### Decision · Program_Chairs · 2026-01-26

Reject